# Can occupational therapist-led home environmental assessment prevent falls in older people? A modified cohort randomised controlled trial protocol

Sarah Cockayne,[1] Alison Pighills,[2] Joy Adamson,[3] Caroline Fairhurst,[1] Avril Drummond,[4] Catherine Hewitt,[1] Sara Rodgers,[1] Sarah J Ronaldson,[1] Sarah E Lamb,[5] Shelley Crossland,[6] Sophie Boyes,[7] Simon Gilbody,[1] Clare Relton,[8] David J Torgerson,[1] on behalf of the OTIS study

For numbered affiliations see end of article.

**Correspondence to**
Mrs Sarah Cockayne;
sarah.cockayne@york.ac.uk

## ABSTRACT

**Introduction** Falls and fall-related injuries are a serious cause of morbidity and cost to society. Environmental hazards are implicated as a major contributor to falls among older people. A recent Cochrane review found an environmental assessment, undertaken by an occupational therapist, to be an effective approach to reducing falls. However, none of the trials included a cost-effectiveness evaluation in the UK setting. This protocol describes a large multicentre trial investigating the clinical and cost-effectiveness of environmental assessment and modification within the home with the aim of preventing falls in older people.

**Methods and analysis** A two-arm, modified cohort randomised controlled trial, conducted within England, with 1299 community-dwelling participants aged 65 years and above, who are at an increased risk of falls. Participants will be randomised 2:1 to receive either usual care or home assessment and modification. The primary outcome is rate of falls (falls/person/time) over 12 months assessed by monthly patient self-report falls calendars. Secondary self-reported outcome measures include: the proportion of single and multiple fallers, time to first fall over a 12-month period, quality of life (EuroQoL EQ-5D-5L) and health service utilisation at 4, 8 and 12 months. A nested qualitative study will examine the feasibility of providing the intervention and explore barriers, facilitators, workload implications and readiness to employ these interventions into routine practice. An economic evaluation will assess value for money in terms of cost per fall averted.

**Ethics and dissemination** This study protocol (including the original application and subsequent amendments) received a favourable ethical opinion from National Health Service West of Scotland REC 3. The trial results will be published in peer-reviewed journals and at conference presentations. A summary of the findings will be sent to participants.

**Trial registration number** ISRCTN22202133; Pre-results.

## Strengths and limitations of this study

- ► The largest randomised controlled trial in an English setting to assess the clinical and cost effectiveness of a home environmental assessment and modification for falls prevention.
- ► Investigates the feasibility of recruiting participants from databases of participants previously assembled while conducting cohort, falls prevention randomised controlled trials.
- ► Hosts three 'Studies within a Trial', which will add to the evidence base about recruitment strategies and ways to minimise missing data within trials.
- ► Results will be generalisable to a community-dwelling population of older people within England.
- ► Uses an unblinded, patient self-report primary outcome measure; therefore, there is a possibility of reporting bias.

## INTRODUCTION

Falls in older people are common and can have serious consequences. Approximately 30% of people over the age of 65 years living in the community will have a fall each year.[1 2] Around 85% of falls occur in the home.[3] One-fifth of all falls are serious and require medical attention with 5% leading to a fracture.[4] Fall-related fractures are a serious cause of morbidity and cost to society.[5] Repeated falls commonly precipitate admission to institutional care and tend to be experienced by frail people in the older age range of 75 years and over.[1 6] The number of falls is likely to increase due to an ageing population and will have a major impact on healthcare resource use, primarily due to hip fractures resulting from a fall. The importance of fall-related injuries has been recognised in the National Service Framework (NSF) for Older People[7] and in the National Institute for Health and Care Excellence (NICE) Guidelines.[8] The NSF calls for

health improvement plans to be devised that will reduce the burden of fall-related injuries.

It is well recognised that many falls result from an interaction between environmental hazards and a broad array of medical conditions and physiological impairments.[9] Environmental hazards are attributed by older people as primary factors in their falls and, thus, frequently cited in the literature as major contributors to falls. 'Accident/environment'related factors were identified as the primary cause of just under one-third of falls in a review of 12 studies (mean of 31%, range 1%–53%, n=3628).[6] Talbot *et al*[10] conducted a retrospective study and identified that 'accident/environment'related factors were perceived by older people as the second most common cause of falls, with key environmental contributors identified as objects on floors, external forces and wet, uneven and icy surfaces.

The theoretical approach underpinning environmental assessment and modification is the person–environment–occupation (PEO) conceptual model of occupational therapy practice.[11] This model posits that the person, environment and task being performed continually interact in ways that enhance or diminish a person's occupational performance and that environmental hazards are dynamic entities that occur through the interaction between these three elements. The PEO model underpins occupational therapy practice that aims to maintain, restore or create a balance between these elements.[12]

The latest Cochrane review in this area (updated September 2012)[13] found that environmental assessment and modification was an effective approach to reducing falls (relative risk of falling 0.88; 95% CI 0.80 to 0.96). It also concluded that the effectiveness of an environmental intervention was increased if delivered by an occupational therapist (OT). Current NICE guidance suggests that 'older people who have received treatment in hospital following a fall should be offered a home hazard assessment and safety intervention/modifications by a suitably trained healthcare professional'. However, at the time of setting up the study, there was no guidance with respect to environmental assessment for people living in the community who are at elevated risk of falling but have not yet received hospital treatment due to a fall. Indeed, there has only been one UK trial of environmental assessment by an OT, which was a pilot study conducted by some of the authors.[14] While this study showed no evidence of a difference between the randomised groups on the primary outcome of fear of falling, a statistically significant reduction was observed in the number of falls (a secondary outcome). Consequently, there is reasonable evidence to suggest OT delivered home hazard assessment and modification can lead to a reduction in falls. This large, multicentre trial builds on this previous work and aims to undertake a high-quality, adequately powered trial to evaluate the clinical and cost-effectiveness of an environmental assessment and modification, delivered by an OT, for the prevention of falls.

## METHODS AND ANALYSIS
### Trial design
The Occupational Therapist Intervention Study (OTIS) study is a modified cohort,[15] pragmatic, two-arm, open, randomised controlled trial (RCT) with an economic evaluation and nested qualitative study. The cohort RCT (cRCT) design was chosen to avoid some of the key potential biases that can occur in a pragmatic trial, namely, high attrition and patient preference effects. In a cRCT, patients are recruited initially into a cohort, and there is usually an outcome run-in period. Given that outcome attrition occurs largely at the first follow-up time point, this attrition is largely avoided if an eligibility criterion for the randomised phase is completion of outcomes during the run-in period.[16] With respect to preference effects, although the control group are aware of the possibility of being offered an intervention (in this case occupational therapy), they are unaware of when the actual randomisation occurs: this might avoid those biases, due to patient preference effects, which relate to timing of the offer of the intervention. In this study, the cRCT design was modified in that both intervention and control groups were told about the intervention prior to randomisation and that which group they would be in would be decided by chance/randomisation. In the cmRCT design, the process of obtaining patient information and consent aims to replicate that in real world routine healthcare, where patients are never told prospectively that their care options will be decided by chance. This approach partly replicates routine care in that the participant is not aware of when randomisation takes place and those in the control group are not aware of when they were formally allocated to be in the comparison group. Similarly, the intervention group are offered the intervention without having to face the possibility that once an offer has been made, that randomisation would withdraw the offer.

### OTIS main study aim
The main aim is to establish whether environmental assessment and modification delivered by an OT will lead to a reduction in the number of falls among those at elevated risk of falling living in the community.

### OTIS secondary aims
This includes:
1. Establishing the cost-effectiveness of OT delivered environmental assessment and modification.
2. Assessing the impact of the intervention on participants' quality of life.
3. Exploring the barriers and facilitators of implementing the trial's findings among OT professionals and the wider community (eg, commissioners of services).

### Participants
#### Participant recruitment
One thousand two hundred and ninety-nine participants will be recruited by one of the following methods (see figure 1).

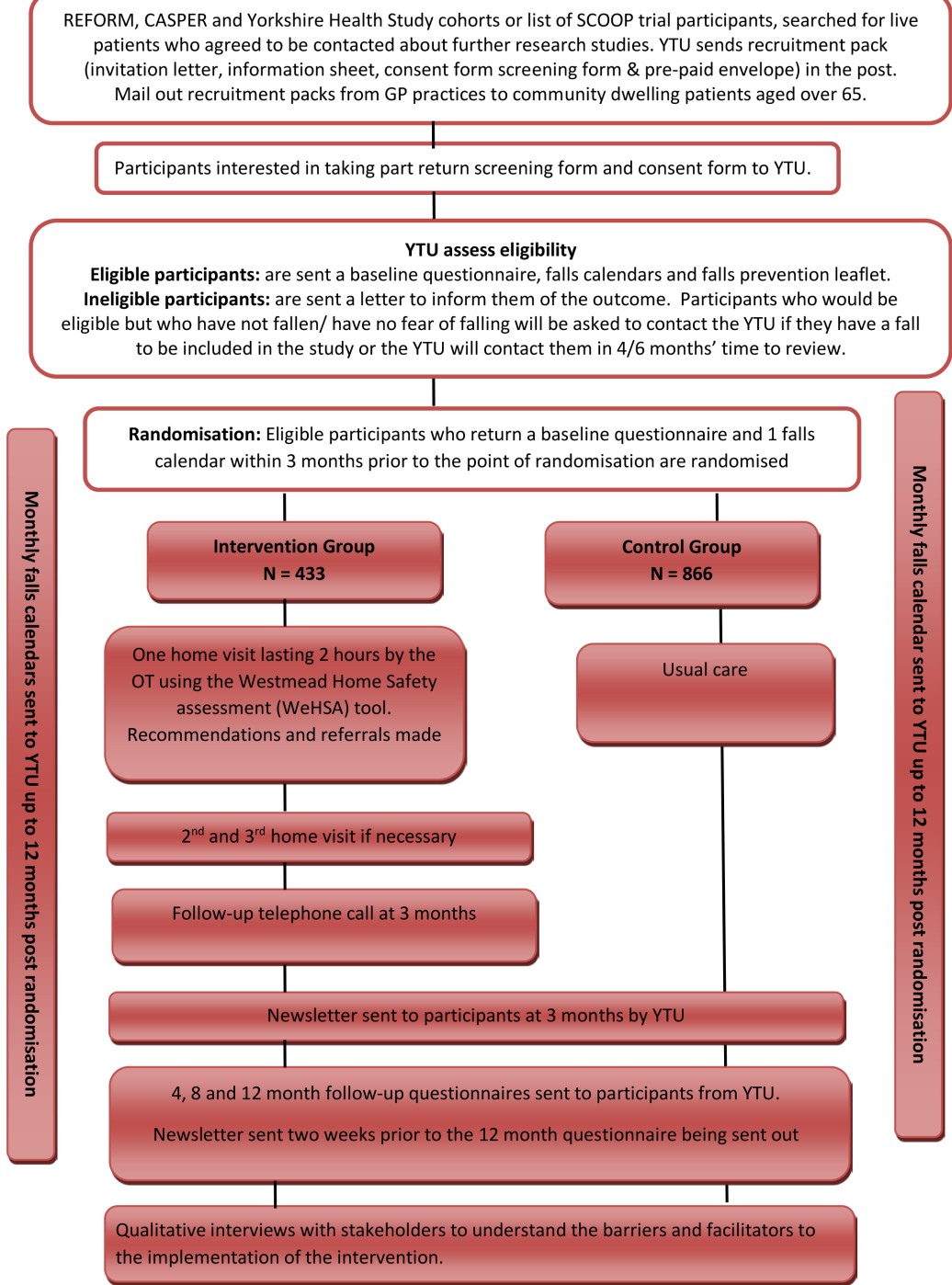

**Figure 1** Flow chart of participants through the OTIS trial. GP, general practitioner; OT, occupational therapist; OTIS, Occupational Therapist Intervention Study; YTU, York Trials Unit.

## Database search of existing cohorts held by the York Trials Unit (YTU) and the Yorkshire Health Study

The YTU has assembled a cohort of participants who originally participated in either the REFORM,[17] SCOOP[18] or CASPER[19] trials and agreed to be contacted about future research studies run by the YTU. These studies recruited participants aged 65 years and over, from either routine National Health Service (NHS) podiatry clinics or GP practices. A database search of these cohorts and the Yorkshire Health Study cohort to identify participants over the age of 65 years[20] will be undertaken to identify participants living in the OTs' catchment area (Yorkshire and North Lincolnshire), who will be eligible for an invitation mailing. Participants known to live in a residential or nursing home will be excluded from the mail out. Potentially eligible participants will be sent an invitation pack asking if they would like to participate in the study. The pack will contain an invitation letter,

participant information sheet, consent form, screening questionnaire and a prepaid envelope. In some cases, the person receiving the invitation pack may decline participation in the study, but a family member or friend may be interested in taking part. In such cases, the original recipient will be asked to pass on the research team's contact details, so that the interested person can contact the study team.

### GP practices and other services

To increase the generalisability of the study's findings, participants will be recruited through GP practices in primary care. GP practices will be recruited to the study after a member of the study team or the local Clinical Research Network has contacted the practice and explained the study and the participants' involvement. A database search will be undertaken to identify community dwelling men and women over the age of 65 who will be sent a recruitment pack. Patients known to have dementia or Alzheimer's disease or who live in a residential or nursing home will be excluded from the mail out by the use of Read Codes (which are a coded thesaurus of clinical terms) and review of the patient's address.

### Opportunistic screening

Where there is capacity, opportunistic screening by other healthcare professionals (eg, GPs, Rapid Assessment Teams, chronic obstructive pulmonary disease nurses, Heart Failure Nurses, Community Matrons or NHS services (eg, ambulance services)) will take place. Potential participants will be given an invitation pack.

### Advertising for participants

Radio, newspaper, faith magazine, social media or television advertisements may also be used to publicise the study and encourage potential participants to get in touch with the researchers. Additionally, posters or flyers may be placed within the geographical area of recruiting sites in places such as supermarkets, libraries and community centres.

Individuals identified via any of the four approaches described above who wish to take part in the study will be asked to return their completed consent form and screening questionnaire by post to the YTU. Researchers will assess the screening form for participant eligibility according to the study eligibility criteria. Participants deemed to be ineligible will be informed in writing. If the respondent is assessed as being ineligible because they have not had a fall within the past 12 months or do not report a fear of falling, but otherwise fulfil the eligibility criteria, they will be given the option to be rescreened in 4–6 months' time.

All eligible, consenting participants will be asked to complete a baseline questionnaire and monthly falls calendars by post. Participants who return a valid baseline questionnaire and at least one falls calendar will be randomised into the trial. Participants can withdraw from the study at any point. The reason for withdrawal will

not have to be declared; however, if provided, this will be recorded. Participants who do not wish to take part in the main study are not required to return any forms to the YTU.

### Inclusion criteria

Participants will be eligible for the OTIS trial if they:
1. Are aged 65 years or over.
2. Are willing to receive a home visit from an OT.
3. Are community dwelling.
4. Have at least one of the following risk factors for a fall in the next 12 months: either one fall in the past 12 months or report a fear of falling on their screening questionnaire.

### Exclusion criteria

Participants will be ineligible for the OTIS trial if they:
1. Are unable to walk 10 feet today (3.05 m) even with the use of a walking aid.
2. Are unable to give informed consent, for example, due to Alzheimer's disease or dementia.
3. Live in a residential or nursing home.
4. Are unable to read or speak English and have no friend or relative to translate/interpret for them.
5. Have had an OT assessment for falls prevention in the previous 12 months.
6. Are on a waiting list for an occupational therapy assessment.
7. Have not returned at least one completed falls calendar in the 3 months prior to randomisation.

### Randomisation

Participants will be enrolled into the study if they fulfil the eligibility criteria and provide written consent to take part in the study; they will then be randomised to either the intervention or control arm when they have returned a valid baseline questionnaire and at least one falls calendar within 3 months prior to the point of randomisation. Randomisation will be carried out using the YTU secure web-based computer randomisation service based on an allocation sequence generated by an independent data systems manager, who is not involved in the recruitment of participants. Participants will be randomly allocated to either the control group or the intervention group in a 2:1 ratio in favour of the control group (to reduce costs). Up to 12 participants from a particular site will be randomised at a time in a single block according to when sites state they have capacity to undertake intervention appointments and for how many participants. The allocation ratio used may go up to 3:1 in a block if the OTs have reduced capacity to carry out the assessment. The YTU will write to the participant's GP informing them of study participation and to participants who are allocated to the intervention group.

### Sample size

We propose to recruit and randomise 1299 participants to the OTIS trial in a 2:1 ratio (ie, 866 to usual care and 433 to intervention). This number allows for 10% attrition and

provides 90% power (using two-sided significance at the 5% level) to show a difference in the percentage of participants who experience at least one fall in the 12 months following randomisation from 60% in the control group to 50% in the intervention group, accounting for the unequal randomisation (StataCorp. 2013. Stata Statistical Software: Release V.13). In the REFORM trial, previously conducted by some of the authors, an absolute difference of 5% was observed in the percentage of participants experiencing a fall (intervention group: 50%; control group: 55%), with a lower confidence limit of 13%; therefore, the decision was made to power this trial for a 10% absolute difference. In the event that sites are struggling with capacity to undertake assessments, we will consider using an allocation ratio of 3:1 (usual care to intervention) to reduce the number of participants they would have to see. If the final ratio was 3:1 (ie, 974 to the usual care and 325 to the intervention), we would have 85% power under the same conditions. The primary outcome is actually a count variable (number of falls, while proportion of participants experiencing at least one fall over the 12 months is a key secondary outcome); however, powering a trial for count data is more complex and requires greater assumptions and so a binary approach to the sample size calculation has been taken here.

### Blinding

Control participants will be blinded to when the intervention takes place; however, due to the nature of the intervention, participants in the intervention group will not be blind. It is also not possible to blind members of the research team who are actively involved in the administration of the study, the statistician or health economist. Data entry staff will be blind to group allocation.

### TRIAL INTERVENTION
### OTIS trial usual care group

Participants will receive usual care from their general practitioner (GP) and other healthcare professionals, which may include referrals to a falls clinic. Participants will receive a falls prevention advice leaflet produced by Age UK ('Staying steady' published in June 2015) with their baseline questionnaire in the post. A group-specific newsletter will be sent to participants at 3 months postrandomisation and 2 weeks before their 12-month follow-up questionnaire is due, informing them about study progress. All participants will receive a pen and £5 with their 12-month follow-up questionnaire in recognition of their participation and to offset any incidental expenses associated with completing the questionnaires.

### OTIS trial intervention group

In addition to the usual care and falls prevention leaflet described above, participants allocated to the intervention arm will be offered a home environmental assessment to identify personal fall-related hazards and behaviours. The assessment will be undertaken by a Health and Care Professions Council registered OT and will take approximately 2 hours to conduct. If the assessment is too demanding for the participant, the appointment may be split over two visits. OTs will attend a 1-day face-to-face training session on how to conduct the assessment. This will be provided by either the researcher who carried out the pilot trial (AP) or two of the OT researchers (ShC and AD) who will be trained by AP to deliver the training in a standardised way.

The environmental assessment will begin with an initial discussion about the participant's history of falling, lifestyle, patterns of usage of areas in the home, risk-taking behaviour, strategies already adopted to reduce falls, environmental changes already in place prior to the assessment and functional vision. This will then be followed by the Timed Up and Go (TUG) test and an environmental assessment using the Westmead Home Safety (WeHSA) tool.[21] The WeHSA was developed in Australia in 1997 for older adults and consists of a 57-item standardised, valid and reliable checklist of fall hazards in the following domains: internal/external traffic ways, general/indoors, living area, seating, bedroom, toilet, bathroom, kitchen, laundry, mobility aid, footwear, pets, medication management and safety call systems. The OT and the participant will move through the house together, and a functional assessment will be completed. Items on the checklist will be rated as either relevant (ie, deemed to be a hazard) or not relevant (ie, not deemed to be a hazard or not present). The OT will discuss any potential falls hazards identified by either the participant or the OT during the assessment and problem solve with the participant to engage them in identifying possible solutions. A list of recommendations will be agreed. If possible, any identified hazards will be removed. If required, the OT will make referrals to other agencies for equipment or a handyman for other minor modifications. They may also make recommendations for equipment that cannot be provided by social services, such as lightweight step ladders with handles and height adjustable rotary washing lines. In such cases, the OT will liaise with the client or a family member regarding purchase of such equipment. The OT will make a clinical judgement whether an additional home visit is required. Four weeks after the assessment, the OT or member of the OTIS research team will telephone the participant to check adherence to the recommendations.

### Treatment fidelity

Treatment fidelity will be assessed using the following combination of strategies.

### Provider training

A standard face-to-face training package will be used to standardise provider training. Training sessions will be recorded where possible. A checklist will be used to document whether all aspects of the training are covered when provided by different facilitators. OTs will have the option to additionally undertake an online training course.

### Delivery of treatment

An observational study will be undertaken over the course of the trial to assess how the treatment was delivered. An OT who delivered the intervention training will shadow OTs while they visit participants. A checklist will be used to record which elements of the intervention are delivered. We will purposively sample OTs for shadowing to ensure we select a sample of OTs who attended different training sessions and who delivered either several or few assessments. Approximately 10 OTs will be observed. While this is a small number of observations involving approximately half of the OTs delivering the intervention, if a greater number were undertaken, then the observation itself would become part of the intervention. Consent for an additional, observing OT to attend the home visit will be obtained from the participant. Participants will be able to decline the second OT attending the visit at any point during the process and will still be able to receive a home visit. Elements of fidelity will also be included in the qualitative interviews. A similar sampling strategy to that detailed above will be used.

### Demonstration of adherence

In order to demonstrate adherence, completion rates of the individual items on the WeHSA will be summarised. In addition to this, an OT who was involved in teaching the delivery of the intervention will review the WeHSA data collected by the OT for each participant. Delivery of the treatment is tailored to individual participant's clinical need; therefore, assessment and recommendations will not be the same for all participants. However, a checklist will be used to document whether the key elements had been covered during each of the consultations.

## OUTCOME MEASURES
### Primary outcome measure for the OTIS trial

The primary outcome is the number of falls per participant over the 12 months from randomisation. A fall is defined as 'an unexpected event in which the participant comes to rest on the ground, floor or lower level'.[22] Data will be collected via participant self-reported monthly falls calendars, on which participants will be asked to mark the number of falls they have on each day or indicate that they have had no falls that month. An explanation of what the researchers consider to be a fall will be included in the participant information sheet and on the falls calendars. If a participant is uncertain as to whether an event is classed as a fall, then they will be encouraged to ring the research team at the YTU to discuss. Participants who do not return their falls calendar within 10 days of the due date will be either telephoned or sent a letter by the YTU to obtain missing data. Participants will be given a Freephone number to ring during office hours to report any falls as soon as possible after the event and when it is safe and convenient to do so. Participants who ring to report a fall will be asked for further details. Participants who indicate on their falls calendar that they have sustained a fall will be telephoned

by the research team for further information. Information collected during the telephone call will include: cause/reason for fall and consequence of fall, for example, superficial wound (bruising, sprain, cut and abrasions), fractures (including type of fracture) and hospital admissions. Data collected from the 4-month, 8-month and 12-month follow-up questionnaires will include falls data and will be used for those participants who do not return their monthly falls calendars.

### OTIS trial secondary outcomes

All secondary outcomes will be self-reported by the participant and collected via questionnaires at baseline, 4, 8 and 12 months or by monthly falls calendars. They include: proportion of participants reporting at least one fall in the 12 months from randomisation; proportion of participants reporting multiple (two or more) falls in the 12 months from randomisation; time to first fall from date of randomisation; health-related quality of life as measured by the EQ-5D-5L[23]; fracture rate; fear of falling as measured by the question 'During the past 4 weeks have you worried about having a fall?'; and health service utilisation.

### Nested qualitative study

To inform potential large-scale implementation of occupational therapy environmental assessment, qualitative interviews will take place with key stakeholder groups involved in intervention delivery (OTs and those who have clinical lead/practitioner roles for falls prevention services). Data will be collected on the feasibility of routinely providing this intervention, barriers and facilitators to implementation, workload implications and readiness to employ this intervention into their regular falls prevention practice. Normalisation Process Theory[24] will be used to guide data collection and to frame the analysis to understand how easy it is to implement these interventions into routine practice.

Fifteen OTs delivering the intervention in the trial and 10 clinical leads who run falls prevention services/care of older people services from organisations involved in the trial and five external to the trial will be purposively selected. Participants will be invited to attend a telephone interview.

### Adverse events

This study will record and report details of any adverse events (AEs) that are required to be reported to the Health Research Authority, that is, events that are related to taking part in the study and are unexpected. The AE reporting period begins as soon as the participant consents to be in the study and ends 12 months after they are randomised.

Details of any AEs will be recorded using a trial adverse event form. Serious adverse events (SAEs) reported by the OT should be reported within 48 hours of the OT becoming aware of the event or within 14 days for non-serious events. A follow-up report will be completed if additional information becomes available.

For this trial, an SAE is defined as any untoward occurrence that:
a. Results in death.

b. Is life threatening.

c. Requires hospitalisation or prolongation of existing hospitalisation.

d. Consists of a congenital anomaly or birth defect.

e. Is otherwise considered medically significant by the investigator.

An event is defined as 'related' if the event was due to the administration of any research procedure. Whereas an 'unexpected event' is defined as a type of event not listed in the protocol as an expected occurrence.

The relatedness of an event will be reviewed by the chief investigator and the Trial Steering Committee. Incidents of hospitalisations, disabling/incapacitating/life-threatening conditions, ageing-associated diseases (such as cancer, cardiovascular disease, diabetes, arthritis, osteoporosis and dementia) and other common illnesses such as depression, falls and deaths are expected in the study population due to the age of the cohort. Similarly, any hospitalisation that was planned prior to entry into the study or cannot be attributed to taking part in the study or prolongation of an existing hospitalisation due to social reasons will not be recorded as an SAE.

## STATISTICAL ANALYSIS

There are no planned interim analyses; therefore, the statistical analysis will be undertaken at the end of the trial and will be conducted using STATA V.15 or later. All analyses will be conducted on an intention-to-treat (ITT) basis, including all randomised patients in the groups to which they were originally allocated. Participant baseline data will be summarised descriptively by group, for all those who have been randomised and for all those who are included in the primary outcome analysis by randomised arm. No formal statistical comparisons will be undertaken. Continuous measures will be reported as means and SD, while the categorical data will be reported as counts and percentages.

### Statistical analysis of the OTIS primary outcome

The number of falls per person will be analysed using Poisson regression (or negative binomial regression, as appropriate) adjusting for gender, age, history of falling and the allocation ratio used to randomise the participant as fixed effects. The model will include an exposure variable for the number of months that the participant returned a monthly falls calendar. A sensitivity analysis will be conducted to account for potential clustering effects by the OT by assigning every randomised participant an OT irrespective of group allocation. For intervention participants, this will be the OT delivering their intervention, whereas for control participants, a counterfactual therapist, that is, one that they could have seen had they been randomised to the intervention group, will be randomly assigned to them. Therapist will then be included as a random effect in the primary analysis model. Additionally, a Complier Average Causal Effect analysis to assess the impact of compliance on treatment estimates will be undertaken for the primary analysis.

### Secondary analysis

The following outcomes will be analysed by logistic regression adjusted for the same covariates as the primary analysis model: the proportion of participants who fall at least once over the 12-month period from the date of randomisation; the proportion of multiple fallers (two or more falls in the 12 months from randomisation); the proportion of participants having at least one fracture over the 12-month follow-up; the proportion of patients obtaining multiple fractures (from different events, if this occurs a sufficient number of times); and the proportion of participants who report that they are worried about falling at 12-month postrandomisation.

Fear of falling will also be analysed in its continuous form using a covariance pattern model incorporating all postrandomisation time points in the analysis and adjusting for baseline score, gender, age, history of falling, allocation ratio, treatment group, time and a treatment group-by-time interaction. The correlation of observations within patients over time will be modelled.

The time to the first fall will be derived as the number of days from randomisation until the patient reports having a fall as detailed in the participant's falls calendars. Time between any subsequent falls will also be calculated. Participants who have not had a fall will be treated as censored at their date of trial exit, or date of last available assessment or 365 days/trial cessation, as appropriate. The proportion of patients yet to experience a fall will be summarised by a Kaplan-Meier survival curve for each group. Time to fall will be analysed using the Andersen and Gill method for analysing time to event data when the event can be repeated. The analysis treats each time to event or censoring as a separate observation. The data will be analysed by Cox proportional hazards regression using robust SEs to account for dependent observations by participant and adjusting for the same covariates as in the primary analysis model.

Adherence to the WeHSA and results of the TUG will also be summarised descriptively.

### Subgroup analysis

The primary analysis will be repeated including an interaction term between the treatment allocation and whether or not a patient received care in a hospital (outpatient appointment, day case, Accident and Emergency presentation or hospital admission) as a result of a fall in the 4 months prior to completion of the baseline questionnaire.

### Missing data

The amount of missing data will be reported by trial arm. A comparison of the baseline characteristics of participants who are included in the primary analysis will be undertaken to ensure that any attrition has not produced imbalance in the groups in important baseline covariates.

A logistic regression model will be used to predict non-response (no falls data received postrandomisation) including all variables collected prior to randomisation. The primary analysis will then be repeated, including as covariates all variables found to be significantly predictive of non-response, to determine if these affect the parameter estimates and study conclusions.

## Qualitative analysis

All interviews will be audio recorded digitally and transcribed verbatim. Initially, following familiarisation with the data, the interview material will be organised according to analytical headings using a constant comparison approach.[25] Key themes will be identified that will then be contextualised in relation to the broader dataset and will be used to assist the interpretation of the trial result. For example, if the intervention is shown to be effective, we will use an Normalization Process Theory (NPT framework to facilitate the development of an implementation plan for integration of occupational therapy falls environmental assessment into routine practice. During the analysis, regular meetings will be held between the qualitative research team and project steering group to discuss emergent themes.

## AE data

AE data will be summarised descriptively by randomised arm.

## Trial monitoring

A Trial Steering Committee and Data Monitoring and Ethics Committee will monitor the trial at least every 12 months or more frequently if the committee requests. The role of this committee will include the review of all SAEs that are thought to be treatment related and unexpected.

## Economic evaluation

The health economic evaluation aims to establish the cost-effectiveness of OT delivered environmental assessment and modification in terms of preventing falls and assess the impact of the intervention on participants' quality of life. The economic analysis will be performed using individual patient level data on an ITT basis. The analytical approach will take the form of cost-effectiveness and cost–utility analyses. The cost-effectiveness approach will assess value for money in terms of cost per fall averted, and the cost–utility analysis will assess cost per quality-adjusted life-year (QALY) gained. The perspective for both analyses will be that of the UK NHS and personal social services, as well as secondary analyses undertaken from a societal perspective. Discounting for future cost and health benefit will not be undertaken given the time frame for the trial is 12 months after randomisation. The year of pricing will be set as the mid-year of the trial.

Health benefits associated with the treatments will be measured in terms of both estimates of the mean number of falls, corresponding to the main outcome of the trial, and mean QALYs, defined as a year lived with full health. In line with NICE recommendations,[26] the EuroQol EQ-5D[27] will be used to elicit patient utility values at different points in time and used to calculate QALYs for each patient, using the area under the curve approach.[28 29] These utility values are used as 'quality adjustment' for each patient's survival time. Specifically, the EuroQol EQ-5D-5L will be used.

Cost data will be collected for each patient regarding healthcare resource use, specifically within primary care and the community (ie, GP, nurse, physiotherapist and OT visits) and the hospital setting (ie, outpatient attendances, day cases, inpatient stays and accident and emergency attendances). Unit costs will then be applied to estimate the total cost per patient. Additional information will be collected regarding intervention costs and private/personal expenses that feed into the societal perspective analysis (eg, activities of daily living equipment and travel costs for healthcare attendances). Unit costs will be obtained from established costing sources such as NHS Reference Costs[30] and PSSRU Unit Costs of Health and Social Care.[31] Data on the cost and utility measures will be collected prospectively at baseline, 4, 8 and 12 months via self-reported questionnaires.

Mean within-trial estimates of cost and health benefits will be estimated using regression methods, allowing for the correlation between costs and effects, as well as adjusting for covariates. The results will be presented as incremental cost-effectiveness ratios, where the difference in mean cost estimates between the two arms is divided by the difference in mean health benefit between the two arms. Findings will also be presented in terms of net health benefit.[32] Multiple imputation methods will be used to handle missing data where needed.[33]

The uncertainty surrounding the decision to accept a treatment as the most cost-effective will be explored in cost-effectiveness acceptability curves (CEACs).[34] These curves depict the probability of accepting a treatment as being cost-effective for a large range of willingness to pay values for an extra unit of health benefit. Sensitivity analysis will be conducted to explore the impact of underlying assumptions of the analysis and the range of unit costs on the cost-effectiveness results.

The main outcome of the trial, falls reduction, is associated with a reduction in fractures. However, due to the restriction in the length of follow-up, the long-term effect in terms of the decreasing number of fractures might not be observed in the current trial. Therefore, a further analysis will explore the possible long-term impact of the trial, assuming that a falls reduction should also lead to a fracture reduction. A decision analytic model approach will be adopted to perform this task. The perspective will be the UK NHS and personal social services, with a lifetime time horizon whereby every participant in a hypothetical cohort is followed up until the last participant dies. The hypothetical cohort will be constructed, based on the characteristics of the trial population, to estimate the QALY yield and cost saving of the long-term effect of the intervention. The model parameters that are not

collected in the trial will be extracted from the existing literature.

The model outputs will be the estimated expected mean costs, effectiveness and QALYs associated with each alternative treatment. Estimated total costs and outcomes will be discounted according to the latest health technology appraisal guidance.[26] Uncertainty regarding cost-effectiveness will be evaluated using probabilistic sensitivity analysis, where inputs into the analysis are defined as probability distributions that reflect uncertainty.[35] The uncertainty surrounding the decision to adopt a given treatment option as a cost-effective treatment at different levels of willingness to pay will be represented in CEACs. The impact of assumptions undertaken in the analysis regarding the evidence over parameters or relating to the decision model (such as extrapolation) will be evaluated in sensitivity analysis, if possible.

## PATIENT AND PUBLIC INVOLVEMENT STATEMENT

Our patient representatives were identified from the cohort of participants who have taken part in previous studies run the members of the study team. They helped develop the design and conduct of the study by providing feedback on the grant application submitted to the funder. We have set up a Patient Involvement Group. This group gives advice to the trial team on the design and conduct of the trial. This included providing input into case report forms, information sheets, participant newsletters and recruitment strategies. They have agreed to help us disseminate our research findings by providing assistance with writing the plain language summaries and the research study findings letter we will send to participants who request it. A member of the group is also a member of the Trial Steering/Data Monitoring and Ethics Committee, where PPI is a standing item.

### Studies within trials

In addition to the main OTIS study, three 'Studies within a Trial' (SWATS) are being conducted.

### Pen substudy

The aim of this substudy is to evaluate the effectiveness of including a pen with the trial invitation pack on recruitment of participants to the OTIS study. Any patient identified in the GP mail out as eligible to receive an OTIS trial invitation pack will be entered into the pen substudy. Block randomisation will be used to allocate participants in a 2:1 ratio in favour of the control group. Generation of the allocation sequence will be undertaken independently by a researcher not involved with the production of the recruitment packs. A single block the size of the number of participants from each GP practice will be used. The intervention group will receive a pen with the YTU logo/details on it; the control group will not receive a pen at the point of being invited to take part in the study. The primary outcome is the proportion of participants who go on to be randomised to the OTIS trial. Secondary outcomes include: the proportion of participants who return a screening form; time to

return screening form; the proportion of participants who fulfil the eligibility criteria apart from the criterion relating to falls within past 12 months or fear of falling; the proportion of participants who are eligible for randomisation; and the proportion of participants who remain in the trial at 3 months postrandomisation. Categorical data will be compared using logistic regression and time to response via a Cox proportional hazards model.

### Invitation letter substudy

The aim of this substudy is to evaluate the effectiveness of writing the potential participant's name by hand on the invitation letter, versus printing their name, on the recruitment rate to the study. Participants will be eligible for this substudy if are they due to be sent an invitation pack about the OTIS trial in the first mail out undertaken by the Yorkshire Health Study. Block randomisation will be used to allocate participants in a 1:1 ratio to receive either a hand written name on the invitation letter (intervention group) or printed name on the invitation letter (control group). Generation of the allocation sequence will be undertaken independently by a researcher not involved with the production of the recruitment packs. The primary outcome is the proportion of participants who go on to be randomised to the OTIS trial. Secondary outcomes include: the proportion of participants who return a screening form; time to return screening form; the proportion of participants who fulfil the eligibility criteria apart from the criterion relating to falls within past 12 months or fear of falling; the proportion of participants who are eligible for randomisation; and the proportion of participants who remain in the trial at 3 months postrandomisation. Categorical data will be compared using logistic regression and time to response via a Cox proportional hazards model.

### Text message substudy

The aim of this substudy is to evaluate the effectiveness of a personalised text message compared with a standard text message on postal questionnaire response rates. Participants who are due to be sent their 4-month follow-up questionnaire and who have provided a mobile phone number and consented to be contacted by text message will be randomised. Block randomisation will be used to allocate participants in a 1:1 ratio to receive either a personalised (intervention group) or a standard text message (control group) at the same time as they are due to receive their postal follow-up questionnaire (ie, 2–4 days after the questionnaire is sent). The randomisation will be stratified by main trial allocation. Generation of the allocation sequence will be undertaken independently by a researcher not involved with the delivery of the text messages.

The personalised text message will read, '*OTIS Trial: [Title, surname of participant] you should have received a questionnaire in the post by now. Your answers are important; so please help by returning it as soon as you can. Thanks.*'.

The standard text message will read '*OTIS Trial: you should have received a questionnaire in the post by now. Your answers are important; so please help by returning it as soon as*

*you can. Thanks.*'. The primary outcome is the proportion of participants in each group who return the questionnaire. Secondary outcomes include time to response, completeness of response, whether a reminder notice is required and cost-effectiveness. Categorical data will be compared using logistic regression and time to response via a Cox proportional hazards model. All models will adjust for main trial allocation.

## Sample size for the SWATs
As is usual with an embedded trial within a trial, no formal power calculation will be undertaken for the pen and text message substudies, as the sample size will be constrained by the number of participants available to either mail out to or contact. We will, however, randomise 314 participants who are due to be mailed out by the Yorkshire Health Study an invitation pack about the OTIS trial. This sample size will allow us to detect a 10% difference in the percentage of participants who go on to be randomised (from 10% to 20%) between the two groups at 80% power and a two-sided alpha level of 0.1.

## ETHICS AND DISSEMINATION
### Ethics
All participants will give written informed consent prior to entry to the study. Further consent will be obtained for the qualitative interviews and fidelity observations.

### Dissemination
The results of the study will be disseminated through high-impact peer-reviewed journals through national and international research conferences and occupational therapy-specific journals and newsletters. A short summary of the results will be sent to participants who request this at the end of the trial.

## DISCUSSION
This study uses a modified cRCT design. The authors have previously conducted three cRCTs.[36] Participants in these trials were recruited from either routine NHS podiatry clinics[17] or from general practices.[18 19] All were aged at least 65 years and over and therefore had an elevated risk of falling. One key feature of the cRCT design is the capacity to undertake multiple RCTs over time. A strength to this trial is that during the recruitment phase of the OTIS study, we will be able to test the feasibility of recruiting participants from these cohorts and determine whether it is a quick and cost-effective means to recruit participants. A further strength lies in the fact that in addition to the main OTIS trial, we have taken the opportunity to undertake three SWATs. The results of these studies will add a significant contribution to the body of evidence about strategies to improve recruitment to trials and minimise the amount of missing data.

One potential limitation to the study is that some participants with mild dementia and cognitive impairment may be included in the study. These participants may have a higher risk of falling. We have tried to exclude this group of participants at screening by collecting data on participant's medical history. In addition, if the study team have any concerns about the ability of a participant to provide informed consent or outcome data during the course of the study, then this is discussed with either the participant, a family member (if the participant consents) or with the participant's GP. Nevertheless, it is still possible that some participants with mild dementia and cognitive impairment may be included in the study that we are unaware of. Further limitations include the fact that the study uses unblinded, patient self-report primary outcome measure, so there is the possibility of reporting bias being introduced. Also, the results of the study will be generalisable to a community-dwelling population of older people within England only.

Falls in older people are a major health problem. A recent Cochrane review found environmental assessment, undertaken by an OT, to be an effective approach to reducing falls in older people. As far as we are aware, none of the trials included a cost-effectiveness evaluation within a UK setting. The OTIS protocol aims to evaluate the clinical and cost-effectiveness of an environmental assessment and modification for preventing falls in older people and will be the largest trial to evaluate this intervention in isolation. If the results of this study are found to be positive, then further research could be conducted to investigate whether the intervention could be delivered equally effectively by trained assessors. Alternatively, further research into the intensity of the intervention, that is, whether more home visits are more effective, could be undertaken.

## Trial status
Recruitment and follow-up are in progress. Recruitment to the study began in October 2015 and will continue until approximately summer 2018. Participants will continue to be followed-up until winter 2019.

**Author affiliations**
[1]York Trials Unit, Department of Health Sciences, University of York, York, UK
[2]Mackay Institute of Research and Innovation, Queensland Health, Mackay Australia and James Cook University, Mackay Base Hospital, Townsville, Australia
[3]Institute of Health & Society, Newcastle University, Newcastle, UK
[4]School of Health Sciences, The University of Nottingham, Nottingham, UK
[5]Nuffield Department of Orthopaedics, Rheumatology and Musculoskeletal Sciences, Botnar Research Centre, University of Oxford, Oxford, UK
[6]Community Mental Health Team, Leicestershire Partnership NHS Trust, Leicester, UK
[7]Occupational Therapy Department, York Teaching Hospital NHS Foundation Trust, York, UK
[8]School of Health and Related Research, University of Sheffield, Sheffield, UK

**Acknowledgements** The research team would like to thank the independent members of the Trial Steering/Data Monitoring and Ethics Committee: Professor Roger Francis, Dr Ranjit Lall, Dr Claire Ballinger, Professor Lindy Clemson and Mrs Margaret McCabe for their advice, overseeing the study and reviewing adverse event data. The authors also extend very grateful thanks to the Patient Involvement Group, the study participants, occupational therapists for delivering the intervention and to the Clinical Research Network for assistance with the General Practitioner's mail out of recruitment packs.

**Collaborators** We would like to thank Dr Jennifer McCaffery and Dr Katie Whiteside, who are based at the York Trials Unit, Department Health Sciences,

University of York, York, UK for their assistance with the day-to-day running of the OTIS trial.

**Contributors** DT and AP wrote the original protocol. All authors were either applicants on the NIHR HTA funding application or helped to refine the protocol. SaC wrote the first draft of this article with input from CF. All authors read and approved the final manuscript.

**Funding** This study is funded by the National Institute for Health Research (NIHR) Health Technology Assessment (HTA) Programme (Programme grant number 14/49/149). The University of York is the study sponsor and is legal responsibility for the initiation and management of the study: sponsor representative Mr Michael Barber, Research and Enterprise Directorate, Innovation Office, York Science Park, Heslington, York UK YO10 5DG. SEL was supported by the NIHR Collaboration for Leadership in Applied Health Research and Care (CLAHRC), Oxford Health National Health Service Trust and the NIHR Biomedical Research Unit, Nuffield Orthopaedic Centre, University of Oxford.

**Disclaimer** The views expressed are those of the authors and not necessarily those of the NHS, the NIHR or the Department of Health.

**Competing interests** CH declares membership of the HTA Commissioning board, and SEL declares membership of the HTA Prioritisation board. No other competing interests are declared.

**Patient consent** Not required.

**Ethics approval** This study protocol (version 7, 19.10.2017) including the original application and subsequent amendments received a favourable ethical opinion from NHS West of Scotland REC 3, the University of York, Department of Health Sciences Research Governance and NHS Trust Research and Development approval at recruiting sites.

**Provenance and peer review** Not commissioned; externally peer reviewed.

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
