## [Reviewer comments · BMJ Open]

ARTICLE DETAILS

TITLE (PROVISIONAL)	Can Occupational Therapist led home environmental assessment prevent falls in older people? A modified cohort randomised controlled trial protocol.
AUTHORS	Cockayne, Sarah; Pighills, Alison; Adamson, Joy; Fairhurst, Caroline; Drummond, Avril; Hewitt, Catherine; Rodgers, Sara; Ronaldson, Sarah; Lamb, Sarah; Crossland, Shelley; Boyes, Sophie; Gilbody, Simon; Relton, Clare; Torgerson, David

VERSION 1 – REVIEW

REVIEWER	Barbara Fischer Clement J Zablocki VA Medical Center USA
REVIEW RETURNED	27-Mar-2018

GENERAL COMMENTS	This manuscript presents the protocol of an ongoing study investigating the effects of a home environmental assessment on falls in older adults. Overall, the study is ambitious and well planned, and addresses an important contributor to falls risk. The environmental assessment appears to be comprehensive and offers an important means by which to reduce falls. Additional sub-studies examine the effects of including a pen, printing versus hand writing participants' names, and the use of text messaging on recruitment and response rates. A concern is the monitoring of fidelity. With approximately 433 proposed home visits, the authors plan to sample only approximately 10 occupational therapists for fidelity to treatment delivery. Please comment on the study's ability to ensure that each home visit is receiving similar assessment and recommendations. Additionally, while the exclusion criteria include individuals with dementia who are unable to give informed consent, individuals with mild dementia and mild cognitive impairment are not addressed. Not only is this population at increased risk for falls, but participants with these conditions may not adhere as closely to treatment recommendations. Please discuss how these factors will be addressed. Specific comments: Page 5 line 49: "It is well recognised that most falls result...." Recommend amending this to "many falls..." Page 7 lines 26-28: "aims to replicate real world routine health care- where patients are never told prospectively that their care options will be decided by chance." This statement warrants further explanation.
--

	Page 8, lines 40-43: Patients known to have dementia...will be excluded." By what means will diagnosis be identified? What about patients with mild cognitive impairment? Page 8, lines 48-54: Where there is capacity...." Are specific medical practices being targeted for recruitment? Please address possible recruitment bias and potential to impact validity.
--	---

REVIEWER	Edgar Ramos Vieira Florida International University, USA
REVIEW RETURNED	09-Apr-2018

GENERAL COMMENTS	Relevance: Important topic. Title: Long and convoluted; consider shortening/simplifying it. Abstract: Clear and well written. Introduction: Clear and to the point. Methods: Well described, but it unclear how TUG times will be analyzed and used considering that the interventions are environmental in nature. Also, you could have used the FES-I (falls efficacy scale) to assess fear of falls, but I understand that this is an ongoing study; just mentioning for future ones... Further information on how the qualitative data will be analyzed is needed (e.g. coding by how many people). Discussion: Comment a bit about the anticipated limitations of the study and future directions. Good study!
---

VERSION 1 – AUTHOR RESPONSE

The OTIS study protocol: a modified cohort randomised controlled trial of whether an Occupational Therapist led home environmental assessment and modification can reduce falls among high risk older people?

Response to peer review comments

Reviewer 1

Comment	Response
This manuscript presents the protocol of an ongoing study investigating the effects of a home environmental assessment on falls in older adults. Overall, the study is ambitious and well planned, and addresses an important contributor to falls risk. The environmental assessment appears to be comprehensive and offers an important means by which to reduce falls. Additional sub-studies examine the effects of including a pen, printing versus hand writing	Thank you for these positive comments. We also believe this is an ambitious study which will add to the evidence base on managing falls in the community.

participants' names, and the use of text messaging on recruitment and response rates.	
A concern is the monitoring of fidelity. With approximately 433 proposed home visits, the authors plan to sample only approximately 10 occupational therapists for fidelity to treatment delivery. Please comment on the study's ability to ensure that each home visit is receiving similar assessment and recommendations.	The observational element is only one of several strategies we are using to assess treatment fidelity. The other strategies we are using include  (1) Provider training. A standardised training package is used and a check is undertaken to document whether all aspects of the training are covered, when provided by different facilitators. (2) Delivery of treatment. Part of our qualitative interviews with the occupational therapists delivering the intervention, will include elements of fidelity. (3) Demonstration of adherence. A summary of the completion rates of the individual items on the Westmead will be reported. In addition to this, a check will be undertaken by the Occupational Therapist (OT) who was involved in teaching the delivery of the intervention, to document whether the key elements have been covered during the consultation. Twenty two OTs are delivering the OTIS study and observations will be conducted with approximately half of those delivering the intervention. The study team acknowledge that this is a small number of observations. However concerns were raised that if more observations were undertaken, then the observation process itself would become part of the intervention. If that were the case, we would not be evaluating

	our research question. Delivery of the treatment is tailored to individual participant's clinical need, therefore assessment and recommendations will not be same for all participants. However, in order to determine whether the key elements have been covered at each home visit, the data collected by the OT for all of the home visits will be reviewed, as detailed in the manuscript under strategy number 3, 'demonstration of adherence'. The issue of fidelity was discussed at great length with both the funders and the independent Trial Steering Committee. Both approved the final strategies described in the protocol. We have now included additional information in the manuscript to clarify this.
Additionally, while the exclusion criteria include individuals with dementia who are unable to give informed consent, individuals with mild dementia and mild cognitive impairment are not addressed. Not only is this population at increased risk for falls, but participants with these conditions may not adhere as closely to treatment recommendations. Please discuss how these factors will be addressed.	We aimed to exclude participants with mild dementia and mild cognitive impairment during the screening process by asking participants and/or their GP about their medical history. Participants who have reported a diagnosis of dementia or Alzheimer's disease have been excluded from the study. Our earlier research from studies within a similar population, suggest that some of the people who have these conditions would not have responded to the invitation mail out and so will not have been included in the study. However, some participants with these conditions may be included in the study and some may develop these conditions during the course of the study. If, during the course of the study, the trial team have any concerns about the ability of a

	participant to provide informed consent and complete outcome data, then this is discussed with the participant or if the participant consents, with a family member or their General Practitioner. These participants may then not complete the trial and may drop out of the study. We have included this as a limitation to the study.
Specific comments: Page 5 line 49: "It is well recognised that most falls result..." Recommend amending this to "many falls..."	Thank you for your comment we have changed the wording as suggested.
Page 7 lines 26-28: "aims to replicate real world routine health care where patients are never told prospectively that their care options will be decided by chance." This statement warrants further explanation.	We have added further explanation as requested.
Page 8, lines 40-43: Patients known to have dementia...will be excluded." By what means will diagnosis be identified? What about patients with mild cognitive impairment?	When patient's electronic medical records are searched to identify potential participants to mail out to, practices are able to search and exclude patients that have a documented history of dementia or Alzheimer's disease by the use of 'Read Codes' (which are a coded thesaurus of clinical terms.) We have now clarified this in the manuscript. Potential participants are sent a screening form on which they are asked "do you suffer from either dementia or Alzheimer's disease?" If the potential participant indicates that this is the case, they are excluded from the study. The

	study team monitor the trial documents returned to them. Participants are contacted if they are having difficulty returning forms. If the trial team have any concerns about the ability of a participant to provide informed consent and complete outcome data, then this is discussed with the participant or if the participant consents, with a family member or their General Practitioner. If needed the participant is then withdrawn from the study. We have now included text discussing this point as a limitation to the study.
Page 8, lines 48-54: Where there is capacity...." Are specific medical practices being targeted for recruitment? Please address possible recruitment bias and potential to impact validity.	In order to aid recruitment we used a number of different methods (e.g. mailing out from our cohorts, mailing out from GP practices, posters and advertisements). One of the ways we tried to recruit was by asking clinicians associated with the study or colleagues of the OTs associated with the study to hand out recruitment packs. Unfortunately, this has not been a successful method of recruiting participants. To date, only three participants have been approached using this method to take part in the study, none of which were eligible for the study. Whilst we agree there is the potential for recruitment bias to be introduced, we do not believe this to be the case, so we have not included any detail in this manuscript. However, if the situation changes, we would include this limitation in the discussion section of the final paper.

Reviewer 2

Comment	Response
---------	----------

Relevance: Important topic. Title: Long and convoluted; consider shortening/simplifying it. Abstract: Clear and well written. Introduction: Clear and to the point. Methods: Well described, but it unclear how TUG times will be analysed and used considering that the interventions are environmental in nature Also, you could have used the FES-I (falls efficacy scale) to assess fear of falls, but I understand that this is an ongoing study; just mentioning for future ones...	As suggested we have reduced the length of the title to: Can Occupational Therapist led home environmental assessment prevent falls in older people? A modified cohort randomised controlled trial protocol. Thank you. Thank you. Data for the TUG will be summarised and used to describe these participants. The TUG will not be used in the analysis. We have added additional information to clarify this. Thank you for this comment, we did consider this and will consider this again in future studies.
Further information on how the qualitative data will be analyzed is needed (e.g. coding by how many people).	As requested, we have included additional information about how the qualitative data will be analysed.
Discussion: Comment a bit about the anticipated limitations of the study and future directions.	As suggested we have included additional limitations to the study and text about future research.
FORMATTING AMENDMENTS (if any) Required amendments will be listed here; please include these changes in your revised version:	

1.No Figure Legend

- Please include Figure legends at the end of your main manuscript.

2.Figure File Format

- Please provide another copy of your figures with better qualities and please ensure that Figures are of better quality or not pix-elated when zoom in. NOTE: They can be in TIFF or JPG format and make sure that they have a resolution of at least 300 dpi. Figures in PDF, DOCUMENT, EXCEL and POWER POINT format are not acceptable.

3.Patient and Public Involvement statement

We have implemented an additional requirement to all articles to include 'Patient and Public Involvement statement'

within the main text of your main document. Please refer below for more information regarding this new instruction:

Authors must include a statement in the methods section of the manuscript under the sub-heading 'Patient and Public Involvement'.

This should provide a brief response to the following questions:

How was the development of the research question and outcome measures informed by

Apologies for this omission. We have now included a Figures legends at the end of our main manuscript.

We have now provided a different file format, which we hope is acceptable.

As requested we have included information for this additional section in the manuscript.

patients' priorities, experience, and preferences? How did you involve patients in the design of this study? Were patients involved in the recruitment to and conduct of the study? How will the results be disseminated to study participants? For randomised controlled trials, was the burden of the intervention assessed by patients themselves? Patient advisers should also be thanked in the contributorship statement/acknowledgements. If patients were not involved please state this. 4. Journal format.	We have checked the format of the manuscript and hope this now complies with the journal's requirements.
--	---

VERSION 2 – REVIEW

REVIEWER	Edgar Ramos Vieira Department of Physical Therapy, Florida International University, USA
REVIEW RETURNED	19-Jun-2018
GENERAL COMMENTS	The authors considered my suggestions and addressed the

	comments. Final suggestion - consider the following title (not a required change, just a suggestion in case you like it): “Do environmental changes in older adults' homes reduce falls? An RCT protocol”.
--	---